# Quantitative Risk Assessment of Five Foodborne Viruses in Shellfish Based on Multiplex qPCR

**DOI:** 10.3390/foods12183462

**Published:** 2023-09-17

**Authors:** Zhendi Yu, Zhangkai Xu, Jiang Chen, Lili Chen, Ningbo Liao, Ronghua Zhang, Dongqing Cheng

**Affiliations:** 1School of Medical Technology and Information Engineering, Zhejiang Chinese Medical University, Hangzhou 310053, China; 202111116011021@zcmu.edu.cn (Z.Y.);; 2Department of Clinical Laboratory, Zhejiang Hospital, Hangzhou 310013, China; 3Department of Laboratory Medicine, The First Affiliated Hospital, Zhejiang University School of Medicine, Hangzhou 310003, China; 4Department of Nutrition and Food Safety, Zhejiang Provincial Center for Disease Control and Prevention, Hangzhou 310051, China; 5College of Food Science and Engineering, Jiangxi Agricultural University, Nanchang 330045, China

**Keywords:** risk assessment, foodborne virus, multiplex PCR, shellfish

## Abstract

Foodborne diseases are currently the most critical food safety issue in the world. There are not many hazard identification and exposure assessments for foodborne viruses (Norovirus GI, GII, Hepatitis A Virus, Rotavirus, Adenovirus) in shellfish. Multiplex qPCR for the simultaneous detection of five foodborne viruses was established and used to assess infection risk based on a 1-year pathogenesis study. The sensitivity, specificity and reproducibility of the multiplex qPCR method are consistent with that of conventional qPCR, which saves more time and effort. Overall, 37.86% of shellfish samples had one or more foodborne viruses. Risk assessment formulae and matrices were used to develop risk assessments for different age groups, different seasons and different shellfish. The annual probability of contracting a foodborne virus infection from shellfish is greater than 1.6 × 10^−1^ for all populations, and even for infants aged 0–4 years, it is greater than 1.5 × 10^−2^, which is much higher than the risk thresholds recommended by WHO (10^−6^) and the US EPA (10^−4^). High risk (level IV) is associated with springtime, and medium risk (level III) is associated with Mussel consumption. This study provides a basis for the risk of foodborne viral infections in people of different ages, in different seasons, and by consuming different shellfish.

## 1. Introduction

Foodborne diseases are currently the most critical food safety issue in the world, and according to the World Health Organization (WHO) statistics, about 30% of the population in developed countries suffer from foodborne diseases each year, and the situation in developing countries is even more serious [1]. There are about 600 million cases of foodborne diseases worldwide each year, and the death toll is as high as 420,000, of which 125,000 are children under 5 years of age [2]. On the other hand, most foodborne diseases are caused by microbial contamination. Virus contamination is an important cause of foodborne diseases in the world.

Norovirus GI (NoV GI) and GII (NoV GII), hepatitis A virus (HAV) and rotavirus (RoV) are considered to have important public health significance. NoV is the most important foodborne virus at present due to its low infection dose, wide transmission route, strong environmental resistance, etc., and it is very easy to contaminate fruits, lettuce and shellfish and to cause outbreaks of foodborne diseases. [3,4]. The pollution of HAV in cold-chain transportation foods such as seafood and fresh berries is particularly serious, because HAV can persist in a low-temperature environment and still maintain its infectivity [5,6]. RoV is the main pathogen causing infectious diarrhea in infants and young children. Around 5% of children worldwide die from rotavirus infection each year [7]. RoV contamination was also detected in shellfish, accounting for 13% in Brazil from 2018 to 2020 [8]. Entero-adenovirus (AdV) is one of the important pathogens causing infant diarrhea. Irrigation water is the most common way of foodborne transmission of AdV [9], but in recent years, AdV contamination has been detected in oyster [10] and shellfish [11].

It can be seen that foodborne viruses can cause outbreaks and epidemics of foodborne diseases through contaminated water, shellfish, fruits and vegetables. Although there are few reports on whether multiple food-borne viral contaminants are present in food, mixed contamination of AdV and RoV has been detected in urban tap water [12]. It can be seen that foodborne viral contamination is often not just a single virus, but mixed contamination is also very common.

The use of a comprehensive risk assessment and risk management can be effective in improving the safety of food [13]. Quantitative Microbial Risk Assessments (QMRAs) generate an understanding of the risks associated with edible shellfish by characterizing the pathogen occurrence in shellfish and evaluating how well these pathogens are controlled by short cooking [12]. The pathogen dose that consumers are exposed to in a particular scenario is translated into probabilities of infection and illness. These can be compared against a tolerable disease burden.

Detection technology is the basis of the monitoring, prevention and control of foodborne diseases. However, due to the complexity of the food matrix, the detection rate is not high, and there are unculturable viruses in food, and the number of virus particles may be low. At present, molecular biology methods represented by polymerase-linked reaction (PCR) are widely used in the detection of foodborne viruses in food because of their sensitive, specific and rapid characteristics [14,15]. However, conventional PCR methods can only detect one virus at a time, and mixed contamination of multiple foodborne viruses in food occurs. Therefore, the conventional PCR technology for the detection of foodborne viruses will consume a lot of time and effort.

In this study, a TaqMan probe-based multiplex qPCR detection method was established in order to simultaneously detect Norovirus GI, GII, Hepatitis A virus, Rotavirus and Adenovirus in commercially available shellfish. QMRA was constructed to assess the risk of foodborne viruses in shellfish based on the results of the multiplex qPCR.

## 2. Materials and Methods

### 2.1. PCR Primers and Probes

Five sets of primers and probes for specific viruses were used in this study (Table 1). All of the primer sets targeted a gene for the capsid protein of the relevant virus. A pair of primers was used for detecting NoV GI, generating a 96 bp PCR product [16]. A pair of primers was used for detecting NoV GII, generating an 89 bp PCR product [16]. To detect HAV, a 174 bp PCR product of the VP3/VP1 junction region was generated using the two primers [17]. Primers named RoV-F and RoV-R were used for amplifying RoV, generating an amplicon size of 128 bp [18]. The amplified product of AdV was 132 bp. All primer sets were synthesized by Sangon Biotech (Shanghai, China). And all information for the primers and probes is provided in Table 1.

### 2.2. Experimental Design for the Multiplex qPCR Method

One Step Primescript PCR Kit (Takara, CA, Shiga, Japan) was used as the reaction mixture for multiplex qPCR. The multiplex qPCR reaction system is as follows: 2 × PCR buffer 12.5 μL, Taq enzyme 0.5 μL, reverse transcriptase 0.5 μL, five virus upstream and downstream primers each 0.6 μL (20 μM), five virus probes each 0.3 μL (20 μM), and 5 μL of nucleic acid template, and fill up the volume to 25 μL with RNase-free water. Multiplex qPCR reaction conditions are as follows: 42 °C reverse transcription for 30 min; 95 °C pre-denaturation for 5 min; 95 °C denaturation for 5 s, 55 °C annealing extension for 35 s, a total of 40 cycles. PCR reaction was performed on the ABI 7500 Fast Real-Time PCR system (ABI, Los Angeles, CA, USA).

### 2.3. Sample Collection and Pretreatment

Six common Marine shellfish samples, including razor clam, scallop, clam, oyster, Venerupis and mussel, were collected from the market in Hangzhou, China once a month, from January 2019 to December 2019. And one or more shellfish samples were collected each time. A total of 103 fresh shellfish samples were collected, including 29 in spring (March to May), 30 in summer (June to August), 18 in autumn (September to November) and 26 in winter (December to February). Detailed sampling is provided in Appendix A.

The pretreatment of samples has been carried out according to the method described by Dirks et al. [19]. An amount of 2.0 g of digestive gland was added to 2.0 mL of Triton X-100 eluate (Sangon Biotech, Shanghai, China). After shaking at 120 rpm, at 37 °C for 60 min, the mixture was subjected to heat at 60 °C for 15 min, centrifuged at 10,000× *g* for 30 min at 25 °C, and the supernatant was transferred to a clean tube. Then, 10% PEG 8000 and 0.3 mol/L NaCl were added and precipitated overnight at 4 °C. The next day, the samples were centrifuged at 10,000× *g* for 5 min at 4 °C, and the precipitate was resuspended in 500 μL RNase-free water. Then, 200 μL mixture was pipetted out for viral nucleic acid extraction with QIAamp Viral RNA Mini Kit (QIAGEN, CA, Stockach, Germany) via the instruction manual [20]. Subsequently, detection was performed via the established qPCR method.

### 2.4. Risk Model Framework

The QMRA simulated the quantitative risk assessment of five foodborne viruses, NoV GI, GII, HAV, RoV and AdV in commercially available shellfish from retail to consumer consumption. The risk model framework is shown in Figure 1. In the exposure assessment, the prevalence and distribution of five foodborne viruses in shellfish were mainly considered at the retail stage, taking into account consumer age, frequency of consumption and total amount consumed. After obtaining the exposure assessment parameters, combined with the dose–response model, MATLAB R2016a software was used to obtain the probability of acute gastroenteritis in different populations caused by eating shellfish through 10,000 iterations.

#### 2.4.1. Hazard Identification

Viruses infect the host mainly through the foodborne route, which can be transmitted by virus-contaminated water, food or aerosols of patient’s excreta [21]. Shellfish are important vectors of foodborne viruses [22]. To assess the risk of foodborne virus infection in shellfish, it is necessary to know the dose of virus that can enter the human body with the shellfish to infect the host, which is mainly calculated from shellfish consumption models.

The shellfish consumption model was developed based on the contamination level of shellfish viruses and the per capita intake of shellfish. The average intake of shellfish per capita referred to the GEMS/Food published by WHO and the area where China was located (region G: Southeast Asia). The formula of shellfish consumption model [23,24]:(1)$$Dose=P_{detect}\times D_{meal}\times V_{copies}\times (\left.1-P_{heat}\right.)$$
*P_detect_* indicates the detection rate of each foodborne virus in shellfish, *D_meal_* represents the amounts of shellfish consumed per meal (g), *V_copies_* denotes shellfish viral load (copies/g), and *P_heat_* represents the rate of virus reduction in shellfish after cooking.

#### 2.4.2. Exposure Assessment

In the exposure assessment stage, the foodborne viruses could not proliferate in shellfish, and the commercial shellfish were mainly regarded as the research objects, so the impact of transportation and storage on viruses in the process of production-consumption was not considered, and the detection results of virus contamination in shellfish were directly used for the analysis. Exposure assessment was conducted based on the contamination level of foodborne viruses in commercially available shellfish, combined with factors such as shellfish dietary consumption and cooking. The main data and procedures of exposure assessment are shown in Figure 1: (1) the detection of foodborne viruses in commercially available shellfish, (2) the contamination level of foodborne viruses in shellfish samples, (3) the amount of shellfish intake per person per meal and the frequency of shellfish intake per year, and (4) the effect of cooking on foodborne viruses.

The exposure assessment started from the marketing, and it was based on the detect data of viral contamination in shellfish. The shellfish consumption model (Equation (1)) was used to calculate the viral infection dose. The virus reduction rate in shellfish was represented by *P_heat_* after cooking for 5 min. *P_heat_* = PERT (0.095, 0.198, 0.367). The per capita shellfish intake was 36.2 g per meal, and the frequency of shellfish intake was 18.3 times per year. The details of shellfish intake are shown in Appendix A. Origin 2018 was used to fit the normal distribution of shellfish consumption model calculated results.

#### 2.4.3. *Dose*–Response Assessment

The hazard results of foodborne viruses were usually measured by dose–response models. Meanwhile, the Beta-Poisson model and exponential model were used to estimate foodborne virus infection [25]. The dose–response model used for NoVs risk assessment in this paper refers to the Fractional Poisson model proposed by Lopman et al. [26].
(2)$$P_{inf}(NoV)=P\times \left(1-e^{\frac{-Dose}{\mu }}\right)$$

*P* indicates a NoVs infection parameter of 0.722, *μ* represents an average polymer size of 1106 during NoVs virus infection, and *Dose* refers to the amount of virus ingested.

The dose–response model for HAV risk assessment refers to the Beta-Poisson model applied in the reports by Ruchusatsawat et al. [27] and Sobolik et al. [28].
(3)$$P_{inf}(HAV)=1-{\left(1+\frac{Dose}{{ID}_{50}}\right)}^{-\alpha }$$

*ID*_50_ represents a dose of 186.69 for HAV to infect 50% of the population, *α* indicates an infection constant of 0.374 for HAV, and *Dose* denotes the amount of virus ingested.

The dose–response model for RoV based on the Beta-Poisson model described by Fuzawa et al. [29] and Bortagaray et al. [30].
(4)$$P_{inf}(RoV)=R-R\times {\left(1+\frac{Dose\times \left(2^{\frac{1}{\alpha }}-1\right)}{{ID}_{50}}\right)}^{-\alpha }$$

*ID*_50_ indicates that the dose of RoV infecting 50% of the population is 6.17, *α* represents that the infection constant of RoV is 0.2531, *R* denotes that the infection coefficient of RoV is 0.35, and *Dose* indicates the amount of virus ingested.

The dose–response model for AdV risk assessment combined with the exponential model of Kongprajug et al. [31] and Lanzarini et al. [32].
(5)$$P_{1}=1-exp(-k\times Dose)$$
(6)$$P_{inf}(AdV)=1-{\left(1+\frac{P_{1}}{\eta }\right)}^{-r}$$

*Exp* is the abbreviation for exponential in Equation (5), which represents the exponential function, *k* represents the infection constant of AdV, and *Dose* indicates the amount of virus ingested. *η* and *r* in Equation (6) are the infection parameters of AdV, which are 6.53 and 0.41, respectively.

#### 2.4.4. Risk Characterization

The results of risk description were obtained via Monte Carlo simulation with MATLAB R2016a software. The probability of acute gastroenteritis caused by shellfish was calculated by substituting exposure estimates into a dose–response model with 10,000 iterations of the Monte Carlo model. The probability of acute gastroenteritis caused by shellfish consumption in one year was estimated based on the model reported by Gao et al. [33].
(7)$$P_{inf,yr}={1-\left(1-P_{inf}\right)}^{n}$$

*P_inf_* denotes the probability of acute gastroenteritis caused by shellfish consumption per meal, and *n* indicates the frequency of shellfish consumption per year.

#### 2.4.5. Risk Assessment Ranking

Referring to the hazard classification of food pathogen in the study of Brusa et al. [34], the hazard levels of foodborne viruses were classified as NoV GI, NoV GII, RoV and AdV with mild hazard, and HAV with moderate hazard. The virus infection risk classification matrix (Appendix A) was established by referring to Hernandez-Jover et al. [35].
Scores for the risk of single virus = Matrix column variable score (hazard level) × Matrix row variable score (infection probability)(8)

According to the detection results of five foodborne viruses in shellfish, the overall risk scores of different shellfish were calculated and analyzed by the risk matrix, and the scores of 1–25, 26–50, 51–75 and 76–100 were defined as very low risk (level I), low risk (level II), medium risk (level III) and high risk (level IV), respectively. The risk classification matrix was used to assess the risk of viral acute gastroenteritis caused by shellfish consumption in different seasons and at different ages.

## 3. Results

### 3.1. Establishment of Multiplex qPCR Method

The multiplex qPCR method comprising five primer sets was developed to detect five kinds of pathogenic viruses (Norovirus GI and GII, Hepatitis A virus, Rotavirus and Adenovirus) simultaneously. The plasmid standard of the foodborne virus was constructed to draw the standard curves of multiplex qPCR. And then, the detection limit and repeatability of the developed multiplex qPCR, as evaluated using serially diluted Plasmid standard, was comparable to that of one-step single qPCR. The qPCR with the five primer sets showed specificity for the pathogenic viruses and showed no cross actions between them (Appendix A). The detection limits were both seen at the 10^1^ copies/μL for NoV GI, NoV GII and HAV, and at the 10^2^ copies/μL for RoV and AdV (Appendix A). The coefficients variation in multiplex qPCR was below 2% (Appendix A), which proves that it has good repeatability. The plasmid standard products with different dilutions were tested, and the standard curve is shown in Figure 2.

### 3.2. Quantitative Detection of Foodborne Viruses in Shellfish via Multiplex qPCR

Five foodborne viruses in shellfish were detected via the established multiplex qPCR method, and the detection results were converted into viral copies by the standard curve to measure the level of virus contamination.

Six common species of shellfish were collected during the year, including razor clam, scallop, clam, oyster, Venerupis and mussel. In one year, mussel had the highest detection rate of foodborne virus (55.56%, 10/18), followed by oyster (47.06%, 8/17). The positive rate of virus in other shellfish samples ranged from 26.67% to 35.29%. Multiple viruses were detected simultaneously in razor clam, scallop, oyster and mussel (Figure 3A). Five foodborne viruses were detected simultaneously in spring and winter (Figure 3B), and the positive rates of foodborne viruses were higher in spring (62.07%, 18/29) and winter (42.31%, 11/26). In spring, 24.14% of shellfish samples (7/29) were detected with multiple viruses (Appendix A), while there was no multiple detection in other seasons. Norovirus GI, GII and Rotavirus were often detected at the same time. No virus was detected in 62.14% of shellfish samples (64/103) (Figure 3C). A total of 14.56% samples (15/103) had viral loads below 100 copies/g, 18.45% shellfish samples (19/103) had viral loads between 100 and 1000 copies/g, 2.91% samples (3/103) contained 1000–10,000 copies/g loads viruses, and only small samples (1.94%, 2/103) had more than 10,000 copies/g viral loads.

It indicates that foodborne virus contamination in shellfish samples is within the controllable range, but more attention should be paid to food safety. Water contamination with acute gastroenteritis viruses detected in shellfish samples also remained mostly low across countries and regions. Contamination levels of foodborne viruses detected in shellfish samples also remained mostly low across countries and regions [36].

The exposure dose of foodborne viruses was calculated via the shellfish consumption model. The exposure dose was fitted with the normal distribution curve to obtain the maximum value μ value ($\overset{-}{x}$) and sigma (*s*) based on the normal distribution curve (Figure 4 and Table 2). It was used for dose–response model calculation. The data of AdV were calculated with mean value and standard deviation ($\overset{-}{x}$ + 1.64 s) as a reference, due to the small number of shellfish samples with AdV detected.

### 3.3. Simulation of the Probability of Foodborne Virus Infection by Eating Shellfish

Based on the exposure assessment results and shellfish diet, the dose–response model was used to calculate the average probability of acute gastroenteritis caused by the five foodborne viruses in shellfish consumed per meal. According to the Chinese dietary survey, the intake of male was 38.4 g per meal, 21.1 times per year. Women consumed 34.3 g per meal, 15.6 times per year. The average intake was 36.2 g per meal and 18.3 times per year (Appendix A). A total of 10,000 Monte Carlo simulations were performed to obtain the average probability of infection. The probability per meal was substituted into Equation (7) to obtain the average probability of acute gastroenteritis per year (Table 3) and probability of shellfish infection in different age groups (Table 4 and Table 5).

The probability of contracting foodborne viruses ranged from 9.7 × 10^−3^ to 5.93 × 10^−2^ after eating shellfish. The risk of Norovirus GI was relatively low, and the risk of Rotavirus was relatively high. However, the risk of shellfish infection per meal was higher than the 1 × 10^−4^ and 1 × 10^−6^ recommended by the US Environmental Protection Agency (US EPA, Washington, DC, USA) and WHO (Geneva, Switzerland) [37,38]. These results indicated that there was a certain risk of infection. The foodborne viral infection risk in shellfish consumption was higher than 1.61 × 10^−1^ per year, being more than 1000 times the recommended risk threshold. It indicated that attention should be paid to the cooking time of shellfish to ensure complete inactivation of residual viruses.

According to the shellfish consumption questionnaire, the 0–4 age group consumed 6.6 g per meal at an average of 8.7 times per year, the 5–18 age group consumed 33.2 g per meal at an average of 12.8 times per year, the average shellfish consumption was 48.9 g per meal and 23.8 times per year in the 19–64 age group, and 23.7 g per meal and 13.8 times per year for those older than 65 years. The risk of foodborne virus infection in different age groups was more than 1.70 × 10^−3^, which was more than 10 times the risk threshold recommended by the US EPA and WHO. In particular, children aged 0–4 years and people over 65 years old should be alert to the risk of foodborne virus infection.

### 3.4. Risk Ranking of Viral Infection

Virus infection risk classification matrix was used to classify the risk of different shellfish, eating season and the age of consuming population. It was divided into very low risk (level I), low risk (level II), medium risk (level III) and high risk (level IV).

The risk level of acute gastroenteritis caused by eating Venerupis was level I. The risk levels of razor clams, scallop, clams and oysters were level II. The risk level for viral acute gastroenteritis from mussel consumption is level III (Figure 5A). Venerupis consumption had the lowest risk level of viral acute gastroenteritis, and mussel consumption had the highest risk level. The risk of viral acute gastroenteritis caused by shellfish consumption was level I for all age groups except for the 19–64 age group, which was level II (Figure 5B). People aged 19–64 years should pay more attention to food hygiene when eating shellfish to protect their health. The risk level of viral acute gastroenteritis caused by shellfish consumption in summer and autumn was level I, and in winter, it was level II. However, shellfish consumption was associated with a class IV risk in spring (Figure 5C).

## 4. Discussion

In recent years, the importance of food safety has become increasingly prominent. Viruses play an essential role in the biological hazards of foodborne diseases. NoV, HAV, RoV and AdV are the main foodborne viruses that cause foodborne gastroenteritis [19]. Serious economic burden and health risks are caused by foodborne viruses worldwide [39,40]. People are infected through consumption of contaminated food or water and direct contact with feces of infected patients or contaminated environmental surfaces. Therefore, the establishment of rapid, sensitive and accurate detection methods can enhance the detection of foodborne viruses in food the risk assessment of infected populations, which is beneficial to reduce infections caused by ingested viruses and is of great significance with regard to food safety and public health [41].

In this study, a multiplex qPCR method was designed to detect NoV, HAV, RoV and AdV, and standard curves were used to quantify the virus content. The multiplex qPCR method has reasonable specificity and stability, and the sensitivity is also equivalent to the standard single-plex PCR method. However, in the multiplex qPCR detection, the fluorescence signal of HAV is lower than the standard single-plex PCR. This is caused by increased competition between reagents when multiple templates bind in the same tube [42]. Multiplex qPCR method is currently a common method for monitoring foodborne virus contamination in food due to cost effectiveness, high reproducibility and reliability. It is also an effective method for virus detection in foodborne epidemics.

The contamination of foodborne viruses in shellfish samples sold in Hangzhou was investigated via the multiplex qPCR method. NoV, HAV, RoV and AdV were positive in commercially available shellfish. At the same time, there is mixed contamination of NoV, HAV and RoV. The finding match those observed in earlier studies [43]. Large concomitant outbreaks of acute gastroenteritis emergency visits in adults and food-borne events suspected to be linked to raw shellfish in France between 2019 and 2020 [44]. Patients with multiple viral co-infections have also been identified. Also, in our study, 24.14% of shellfish (7/29) were found to have multiple viruses in spring (Appendix A), while no multiple viruses were detected in other seasons. Both HAV and RoV were detected in one clam and one oyster, NoV GI and RoV were detected in another oyster, NoV GI and NoV GII were detected in two mussels and one scallop, and NoV GII and RoV were detected in one mussel. The detection of multiple viruses in shellfish may be related to the co-infection of multiple viruses in patients. It can be seen that there is still foodborne virus contamination in the currently marketed shellfish seafood, and shellfish seafood is still the primary carrier of foodborne viruses infecting the human body [45]. Foodborne viruses were detected in large quantities of raw shellfish consumed in our study, indicating widespread contamination of the harvesting sites. Previous studies have shown that contamination of shellfish with foodborne viruses is usually associated with various environmental factors, including heavy rainfall prior to harvest [46]. Heavy rainfall can cause sewage treatment plants to overflow, leading to fecal contamination of water [47]. Understanding the causes of widespread contamination in shellfish production areas, in particular, precipitation, sewage treatment plant overflows and fecal contamination of water sources, can help to better control foodborne viral contamination of shellfish.

Quantitative results obtained from QMRA assess the health risk of pathogen exposure and provide a basis for pathogen prevention and control [48]. In this study, the probability of infection from exposure to five foodborne viruses, namely NoV, HAV, RoV and AdV in shellfish was calculated using QMRA. The probability of contracting foodborne viruses from shellfish consumption per meal and per year for different age groups is shown in Table 4. We found that the annual probability of contracting foodborne viruses from shellfish consumption in the population was higher than 1.0 × 10^−1^, which far exceeds the risk thresholds recommended by WHO (10^−6^) and US EPA (10^−4^). According to the study, the probability of NoV infection from oyster consumption in the US and Canada was 0.108 and 0.188, which is similar to our results [43]. Owusu et al. investigated the annual probability of NoV infection in wastewater-irrigated vegetables and showed that the probability of infection ranged from 9.2 × 10^−1^ to 9.4 × 10^−1^ for all genotypes of NoV [49]. Ankita Bhatt et al. found that the probability of infection for AdV and NoV both reached 0.9 [50]. The best option for assessing the current situation and QMRA in a given environment is to compare the values with those recommended by WHO (10^−6^) and US EPA (10^−4^) due to the different types of viruses in many studies. The risk of foodborne viruses in shellfish samples was found to be well above the recommended risk thresholds in our current testing. Cooking time should be observed in consumption to reduce the health risk due to exposure to foodborne viruses. In this study we collected six species of commercially available shellfish throughout the year, but the sample numbers were small, and the reliability of the probabilities obtained from the QMRA was limited. The sampling volume should be increased to improve the reliability of the data.

## 5. Conclusions

In short, we have established a multiplex qPCR method to detect and quantify NoV, HAV, RoV and AdV. Our results show that the established multiplex qPCR method has reasonable specificity and reproducibility, and the sensitivity is equivalent to the single-plex PCR method. On the other hand, we used the established method to monitor the marketed shellfish. NoV, HAV, RoV and AdV were detected in six species of shellfish throughout the year, and individual samples had extremely high loads of virus. The risk of foodborne viral infections from shellfish consumption throughout the year was well above the WHO and US EPA risk thresholds, as assessed by QMRA. High risk (level IV) is associated with spring-time, and medium risk (level III) is associated with mussel consumption.

## Figures and Tables

**Figure 1 foods-12-03462-f001:**
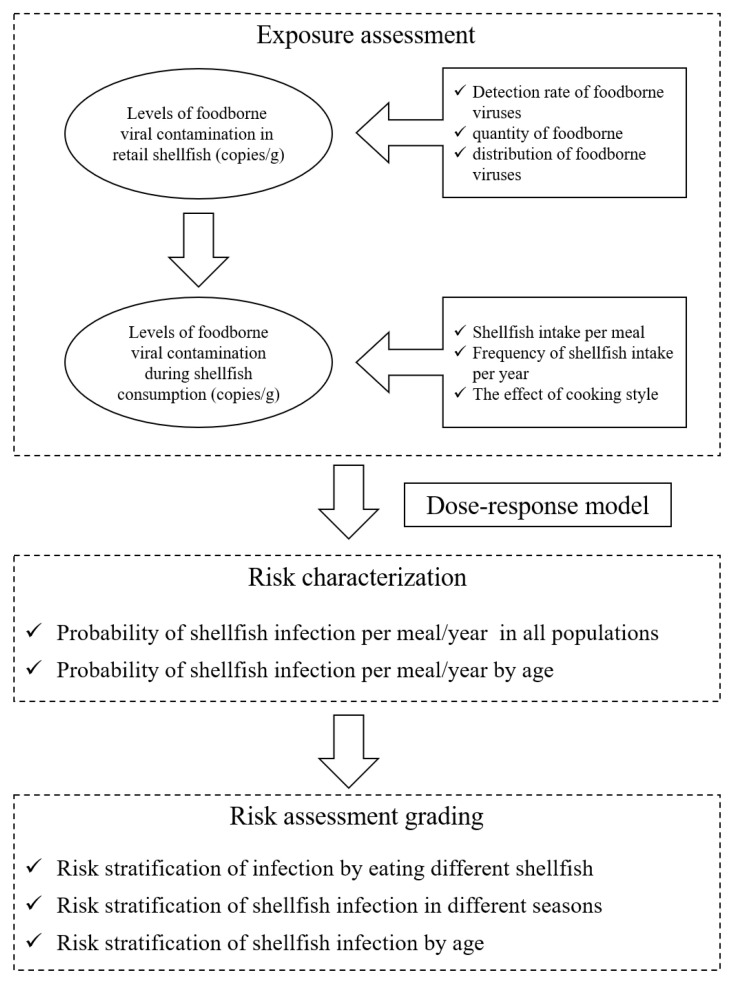
Risk model framework.

**Figure 2 foods-12-03462-f002:**
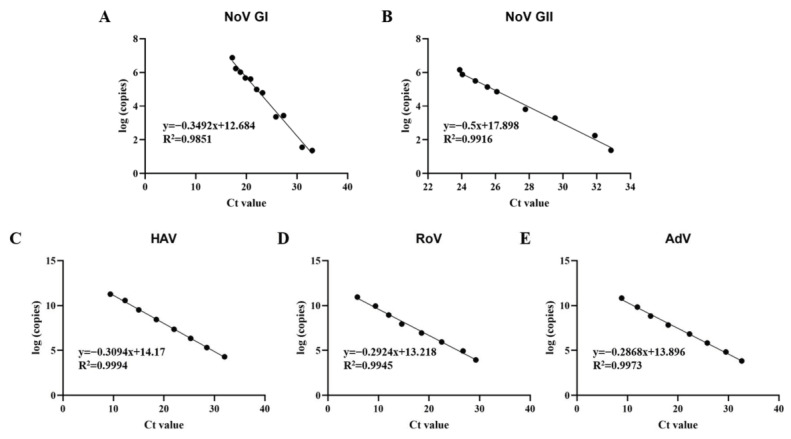
Standard curves of multiplex qPCR assays. (**A**) Standard curve of Norovirus GI; (**B**) Standard curve of Norovirus GII; (**C**) standard curve of Hepatitis A virus; (**D**) standard curve of Rotavirus; (**E**) standard curve of Adenovirus.

**Figure 3 foods-12-03462-f003:**
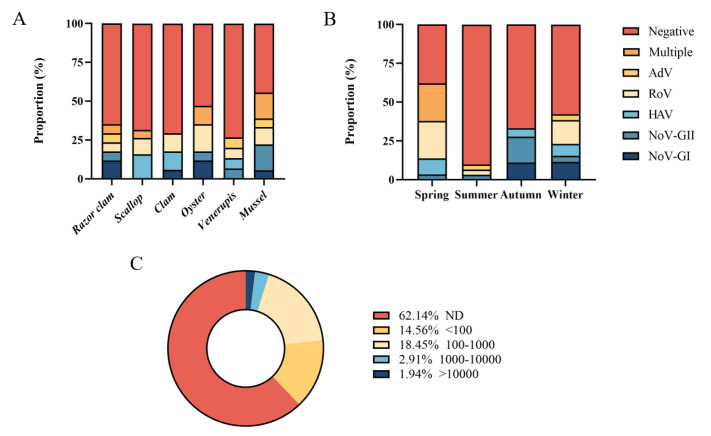
Quantitative detection of foodborne viruses. (**A**) Proportion of different shellfish samples. (**B**) Proportion in different seasons. (**C**) Distribution of samples with different viral copy numbers.

**Figure 4 foods-12-03462-f004:**
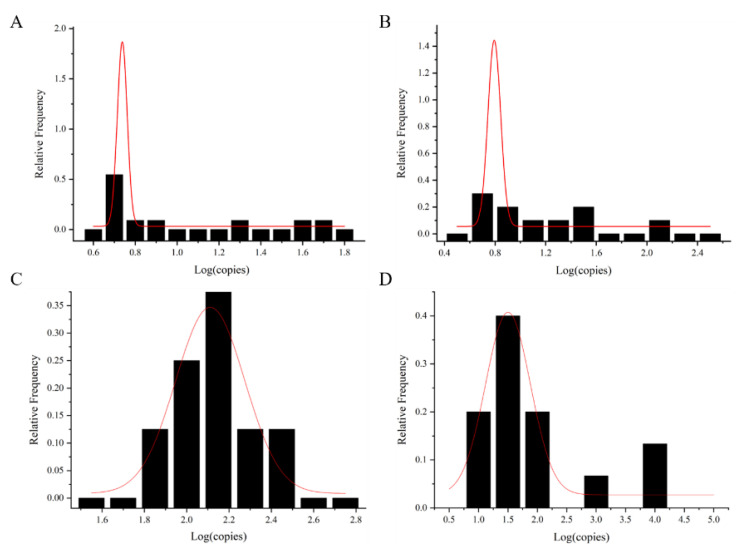
Normal fit distribution of foodborne viral loads in shellfish samples. (**A**) NoV GI, (**B**) NoV GII, (**C**) HAV, and (**D**) RoV. The red line is the normal distribution fitting curve.

**Figure 5 foods-12-03462-f005:**
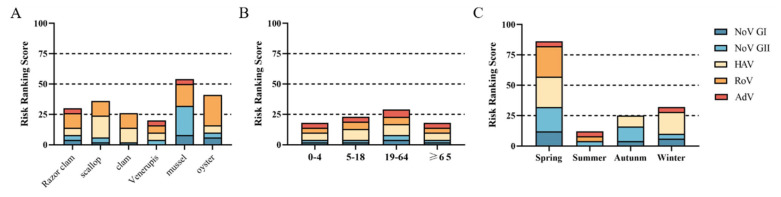
Risk ranking of viral infection. (**A**) Different species of shellfish. (**B**) Different age groups. (**C**) Different seasons.

**Table 1 foods-12-03462-t001:** Primer and probe sequences.

Primer	Sequence (5′-3′)
NoV GI-F	GCCATGTTCCGITGGATG
NoV GI-R	TCCTTAGACGCCATCATCAT
NoV GI-P	HEX-AGATYGCGRCTCCCTGCTCACA-BHQ1
NoV GII-F	CAAGAGTCAATGTTTAGGTGGATGAG
NoV GII-R	TCGACGCCATCTTCATTCACA
NoV GII-P	FAM-AGATTGCGATCGCCCTCCCA-BHQ1
HAV-F	TCACCGCCGTTTGCCTAG
HAV-R	GGAGAGCCCTGGAAGAAAG
HAV-P	ROX-CCTGAACCTGCAGGAATTAA-BHQ2
RoV-F	ATGGATGTCCTGTACTCCTTGTCAAAA
RoV-R	TTCCTCCAGTTTGRAASTCATTTCC
RoV-P	CY5-AATGTACCTTCAACAATYTTRTCCCTAGC-BHQ2
AdV-F	GCCCCAGTGGTCTTACATGCACATC
AdV-R	GCCACGGTGGGGTTTCTAAACTT
AdV-P	CY3-TGCACCAGACCCGGGCTCAGGTACTCCGA-TAMRA

**Table 2 foods-12-03462-t002:** Normally fitted distribution data for foodborne virus content in shellfish.

	NoV GI	NoV GII	HAV	RoV	AdV
$\mu (\overset{-}{x})$	0.7378	0.7934	2.1113	1.5001	2.6975
sigma	0.0237	0.0501	0.1633	0.3812	0.1623

The data of AdV were calculated with mean value and standard deviation ($\overset{-}{x}$ + 1.64 s) as a reference, due to the small number of shellfish samples with AdV detected.

**Table 3 foods-12-03462-t003:** Risk of acute gastroenteritis caused by shellfish consumption.

	Risk of Infection per Meal	Annual Consumption Risk of Infection
NoV GI	9.7 × 10^−3^	1.61 × 10^−1^
NoV GII	1.02 × 10^−2^	1.66 × 10^−1^
HAV	5.41 × 10^−2^	5.14 × 10^−1^
RoV	5.93 × 10^−2^	6.07 × 10^−1^
AdV	4.97 × 10^−2^	5.98 × 10^−1^

**Table 4 foods-12-03462-t004:** Risk of acute gastroenteritis from shellfish consumption per meal in different age groups.

Age	Risk of Infection per Meal				
	NoV GI	NoV GII	HAV	RoV	AdV
0–4	1.70 × 10^−3^	1.90 × 10^−3^	1.15 × 10^−2^	1.82 × 10^−2^	3.39 × 10^−2^
5–18	8.90 × 10^−3^	9.30 × 10^−3^	5.03 × 10^−2^	5.64 × 10^−2^	4.91 × 10^−2^
19–64	1.31 × 10^−2^	1.37 × 10^−2^	6.92 × 10^−2^	6.99 × 10^−2^	5.17 × 10^−2^
≥65	6.40 × 10^−3^	6.70 × 10^−3^	3.76 × 10^−2^	4.60 × 10^−2^	4.64 × 10^−2^

**Table 5 foods-12-03462-t005:** Annual risk of acute gastroenteritis from shellfish consumption in different age groups.

Age	Annual Consumption Risk of Infection				
	NoV GI	NoV GII	HAV	RoV	AdV
0–4	1.54 × 10^−2^	1.61 × 10^−2^	9.22 × 10^−2^	1.42 × 10^−1^	2.49 × 10^−1^
5–18	1.07 × 10^−1^	1.11 × 10^−1^	4.07 × 10^−1^	4.81 × 10^−1^	4.69 × 10^−1^
19–64	2.61 × 10^−1^	2.69 × 10^−1^	6.46 × 10^−1^	7.42 × 10^−1^	7.11 × 10^−1^
≥65	8.37 × 10^−2^	8.72 × 10^−2^	3.52 × 10^−1^	4.37 × 10^−1^	4.71 × 10^−1^

## Data Availability

The data used to support the findings of this study can be made available by the corresponding author upon request.

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
