# Peer review of "Quantitative Risk Assessment of Five Foodborne Viruses in Shellfish Based on Multiplex qPCR"

_foods, 2023, doi:10.3390/foods12183462_

Round 1
Reviewer 1 Report
Table 1 is not provided!?
Page 2, line 92: four or five virus upstream and downstream primers?
Page 3, Subsection 2.3 & related to Table S1: Does the digestive gland samples derived from one mussel? As you have 103 mussels and during the three-month periods you occasionally have more than one mussel sample (each species) per month, and do you collect each mussel samples (six species) on monthly bases or rarely…? I believe that sample collection (and related qPCR results) should be precisely described and explained.
Page 3, line 105: The term "KNT" deserves to be explained.
Page 4, Figure 3.: My suggestion is to remove the negative data of Positive rate (%) in Figures 3A and 3B (in accordance with figure caption). Thereby, the y-axis can extend to e.g. 70%, and the data will be observable. However, legend should be included in both Figures 3A and 3B.
Page 7, lines 254-255: As the Figure 4. didn’t provide the fit distribution of foodborne AdV loads in samples, I will prefer to see this sentence ("The data of AdV…") in main text.
Page 7, Table 4.: I would prefer to see and recommend to split Table 4 in two tables.
Page 8, Subsection 3.4: Instead the "Fig 4A", "Fig 4B", and "Fig 4C", it should be "Fig 5A", "Fig 5B", and "Fig 5C", respectively.
Page 9, Section 4 (related to statement in lines 3276-327): I would like and prefer to see some discussion related to Figure 4 results - multiple detection of foodborne viruses and possible co-infections (e.g. what is the frequency of multiple infections of your samples, how many and which virus species were detected, possible season influence, etc.…).
Author Response
Dear Editor and Reviewer:
Thank you for your letter and for the reviewers' comments concerning our manuscript entitled “Quantitative risk assessment of five foodborne viruses in shellfish based on multiplex qPCR”. Those comments are all valuable and very helpful for revising and improving our paper, as well as the important guiding significance to our research. We have studied comments carefully and have made correction which we hope meet with approval. The main corrections in the paper and the responds to the reviewer's comments are in the appendix.

Reviewer 2 Report
The present study reports the quantitative risk assessment of five foodborne viruses in shellfish based on multiplex qPCR. Overall, the manuscript is well disgned, and could be improved as follow:
-L 87: make "a" capital in and, add dot at the end (L88).
-L98-99: check the shellfish names why italic? this is not scientific name to be italic, moreover, you could write the scientific name in front of each type.
-L 99; specify the country after Hangzhou.
-L 100: what about the storage conditions of these sample? is the analyses conducted immediatly or after 2019??
-L 104; instead of "is" write "has been carried out".
-L 105: methods should be revised and wrote in more detalis, e.g., KNT eluate, all reagents should be wrote with the manufacturers and the country, as well the instruments used with their models.
-L 105: write the details of the shaker and the temperature.
-Convert rpm to xg. Add the centrifuge model, the temperature.
-L 106: write the abbreviation mean of PEG 8000, apply for similar cases.
-Provide Fig. 1 in high quality.
-Indicate the equation numbers for EQ 2-3-4-5 in the text.
-Fig. 2: in the captiion, write the complete meaning of each abbreviated virus, e.g., Norovirus GII (NoV GII), also, apply this issue, in the first time mentioned in the mansucript.
-L 227: check the names.
-Has the statistical anlyses done?
Should be improved.
Author Response

(The authors gave the same response as above.)
